# Indirect SPECT Imaging Evaluation for Possible Nose-to-Brain Drug Delivery Using a Compound with Poor Blood–Brain Barrier Permeability in Mice

**DOI:** 10.3390/pharmaceutics14051026

**Published:** 2022-05-10

**Authors:** Asuka Mizutani, Masato Kobayashi, Makoto Ohuchi, Keita Sasaki, Yuka Muranaka, Yusuke Torikai, Shota Fukakusa, Chie Suzuki, Ryuichi Nishii, Shunji Haruta, Yasuhiro Magata, Keiichi Kawai

**Affiliations:** 1Faculty of Health Sciences, Institute of Medical, Pharmaceutical and Health Sciences, Kanazawa University, 5-11-80 Kodatsuno, Kanazawa 920-1192, Japan; mizutani.a@staff.kanazawa-u.ac.jp (A.M.); kobayasi@mhs.mp.kanazawa-u.ac.jp (M.K.); makohhm359@gmail.com (M.O.); yukarisa93@stu.kanazawa-u.ac.jp (Y.M.); 2R&D Department, TR Company, Shin Nippon Biomedical Laboratories, Ltd., 2438 Miyanoura, Kagoshima 891-1394, Japan; sasaki-keita@snbl.co.jp (K.S.); torikai-yusuke@snbl.co.jp (Y.T.); fukakusa-shota@snbl.co.jp (S.F.); haruta-shunji@snbl.co.jp (S.H.); 3Department of Molecular Imaging, Institute for Medical Photonics Research, Preeminent Medical Photonics Education & Research Center, Hamamatsu University School of Medicine, 1-20-1 Handayama, Higashi-ku, Hamamatsu 431-3192, Japan; csuzuki@hama-med.ac.jp (C.S.); ymagata@hama-med.ac.jp (Y.M.); 4Department of Molecular Imaging and Theranostics, Institute for Quantum Medical Science, Quantum Life and Medical Science Directorate, National Institutes for Quantum Science and Technology, 4-9-1 Anagawa, Inage, Chiba 263-8555, Japan; nishii.ryuichi@qst.go.jp; 5Biomedical Imaging Research Center, University of Fukui, 23-3 Matsuokashimoaizuki, Eiheiji, Yoshida-gun, Fukui 910-1193, Japan

**Keywords:** nose-to-brain drug delivery, poor BBB-permeable drug, brain drug distribution, dopamine D2 receptor antagonist, SPECT, [^123^I]IBZM

## Abstract

Single-photon emission computed tomography (SPECT) imaging using intravenous radioactive ligand administration to indirectly evaluate the time-dependent effect of intranasal drugs with poor blood-brain barrier permeability on brain drug distributions in mice was evaluated. The biodistribution was examined using domperidone, a dopamine D2 receptor ligand, as the model drug, with intranasal administration at 0, 15, or 30 min before intravenous [^123^I]IBZM administration. In the striatum, [^123^I]IBZM accumulation was significantly lower after intranasal (IN) domperidone administration than in controls 15 min after intravenous [^125^I]IBZM administration. [^123^I]IBZM SPECT was acquired with intravenous (IV) or IN domperidone administration 15 min before [^123^I]IBZM, and time–activity curves were obtained. In the striatum, [^123^I]IBZM accumulation was clearly lower in the IN group than in the control and IV groups. Time–activity curves showed no significant difference between the control and IV groups in the striatum, and values were significantly lowest during the first 10 min in the IN group. In the IN group, binding potential and % of receptor occupancy were significantly lower and higher, respectively, compared to the control and IV groups. Thus, brain-migrated domperidone inhibited D2R binding of [^123^I]IBZM. SPECT imaging is suitable for research to indirectly explore nose-to-brain drug delivery and locus-specific biological distribution.

## 1. Introduction

Almost all potential therapies are greatly restricted from entering the brain because the blood-brain barrier (BBB) is highly successful at protecting the brain from the entry of drugs [1,2]. P-glycoprotein (P-gp) is considered one of the most important efflux transporters located at the BBB, and it protects the brain from chemicals and toxins. For this reason, many drugs have an affinity for P-gp and drugs administered intravenously cannot accumulate in the brain due to the BBB. Nose-to-brain (N2B) drug delivery is an attractive route for direct drug delivery from the nose to the brain that bypasses the BBB [3]. Neotrofin, a P-gp substrate, administered intracerebroventricularly in the lateral cerebral ventricle is inhibited from being excreted from the brain when combined with verapamil, a P-gp inhibitor [4]. Thus, P-gp substrate drugs may temporarily accumulate in the brain via N2B drug delivery because they are excreted from the brain when they are recognized by P-gp in the brain. However, it remains unclear whether P-gp-mediated efflux has an effect on time-dependent drug accumulation in the brain. Therefore, it is useful to establish a method for accurate evaluation of the effect of intranasal P-gp substrate drugs.

Radionuclide imaging using positron emission tomography (PET), or single-photon emission computed tomography (SPECT), enables direct, non-invasive measurement of functional proteins (e.g., receptors, transporters) in a living organism. A previous PET study evaluated brain uptake of ^11^C-labeled CH_3_-Orexin A after intranasal delivery in rodents and nonhuman primates. However, these imaging methods have issues that include the technical difficulty of labeling an evaluation compound with a radioisotope, a high likelihood of changes in properties of an evaluation compound by labeling, as well as uncertainties in detecting labeled metabolites rather than a labeled intact (non-labeled) compound. In addition to these issues, if the above method is applied to N2B drug delivery, there is great difficulty in achieving rigorous detection of very slight and sensitive brain uptake of a radiolabeled evaluation compound because of extreme differences in radioactivity between the nasal cavity and the brain [5].

Instead of the above method of direct evaluation of a radiolabeled compound in the brain by PET, indirect evaluation using brain PET imaging of specific receptor occupancy has been performed to examine the time-dependent effect of naloxone administered intranasally on mu opioid receptor occupancy in nonhuman primates and human subjects, with intravenous injection of [^11^C]carfentanil as a PET tracer ligand into the receptor [6,7]. Since the naloxone used in these studies is highly nasally absorbable, BBB-permeable, and does not activate efflux transporter P-gp [8,9], it is possible that the results of receptor occupancy with nasal naloxone in these studies are based on brain naloxone distributions via the BBB route rather than the N2B route. Although brain PET imaging with a ^11^C-labeled compound of receptor occupancy enables the use of an intact evaluation compound and includes evaluation of its pharmacodynamics in the brain, the volume of ^11^C-labeled compounds injected into the subject is usually high because ^11^C-labeled compounds in new radiopharmaceuticals have a very short radioactive half-life and low radioactive concentration. In this study, we demonstrated a SPECT imaging method that uses intravenous radioactive ligand administration to indirectly evaluate the time-dependent effect of intranasal drugs with poor BBB permeability on brain drug distribution in mice. To evaluate brain drug distribution in mice via only the N2B route, the present mouse study used domperidone, a dopamine D2 receptor (D2R) antagonist, which has been reported to have only peripheral effects due to its poor BBB permeability caused by active efflux transport of P-gp [10,11], as a model drug for N2B delivery. In addition, (*S*)-(-)-3-iodo-2-hydroxy-6-methoxy-*N*-[(1-ethyl-2-pyrrolidinyl)-methyl]benzamide ([^123^I]IBZM), a D2R ligand with known high BBB permeability, was selected as the radiopharmaceutical to indirectly evaluate brain domperidone distribution by competing D2R binding using SPECT imaging.

## 2. Materials and Methods

### 2.1. Materials

(*S*)-(-)-3-trimetylstannyl-2-hydroxy-6-methoxy-*N*-[(1-ethyl-2-pyrrolidinyl)methyl]-benzamide (TBZM; >95% purity) and *S*-(-)-IBZM ([^127^I]IBZM, >95% purity) were purchased from the NARD Institute, Ltd. (Hyogo, Japan) and ABX (Radeberg, Germany), respectively. ^125^I-NaI (3.3–4.1 GBq/mL) and ^123^I-NaI (8.5–15.0 GBq/mL) were purchased from PerkinElmer, Inc. (Waltham, MA, USA) and FUJIFILM Toyama Chemical Co., Ltd. (Tokyo, Japan), respectively. Chloramine-T (>95% purity), sodium metabisulfite (>95% purity), and ammonium acetate (>97% purity) were purchased from Nacalai Tesque, Inc. (Kyoto, Japan); domperidone (>90% purity), ethanol (>99.5% purity), and hydrogen carbonate (>95% purity) were provided by FUJIFILM Wako Pure Chemical Corporation (Osaka, Japan).

### 2.2. Labeling of [^125/123^I]IBZM

To synthesize [^125/123^I]IBZM following the method of Kung [12] and the schematic diagram shown in Figure 1, 455 μM TBZM (20 μL) diluted by adding 0.5 M ammonium acetate (100 μL) and ethanol were mixed with ^125^I-NaI or ^123^I-NaI and 0.1 μM chloramine-T (25 μL). The reaction mixture was incubated at room temperature for 5 min. The oxidation reaction was stopped by the addition of 1/10 saturated sodium metabisulfite. Then, pH was adjusted to about 7.0 using 700 μL of 1/2 saturated 0.12 M hydrogen carbonate. The labeling rate of [^125/123^I]IBZM was >80% using a thin layer chromatography, and the radiochemical purity was >95% using high-performance liquid chromatography (HPLC), comprising a pump (model L-7100; Hitachi High-Tech Global, Ibaraki, Japan), a UV detector (Chromaster 5410, Hitachi High-Tech Global), and a γ-ray detector (model RLC-701, Hitachi, Ibaraki, Japan) equipped with a 5C_18_ AR-II-column (4.6 × 250 mm; 5 µm, Nacalai Tesque). The mobile phase consisted of 10% 4 mM ammonium phosphate buffer (pH 7.0) and 90% acetonitrile at a flow rate of 1.0 mL/min. The radioactive concentrations of [^125^I]IBZM and [^123^I]IBZM were approximately 3.7 MBq/mL and 130 MBq/mL, respectively. The spectrum of [^125/123^I]IBZM was identified in comparison with the spectrum of [^127^I]IBZM in the HPLC system (Appendix A).

### 2.3. Animals

All applicable institutional guidelines of Kanazawa University for the care and use of animals were followed. All procedures were performed in accordance with the ethical standards of Kanazawa University (Animal Care Committee of Kanazawa University, AP-173851) and with international standards for animal welfare and institutional guidelines. The mice used in the study (ddY, males, 28.5 ± 0.9 g, 29 mice in total) were obtained from Japan SLC, Tokyo, Japan. Tracheal cannulation according to the method of Hirai et al. [13] was performed in all mice to increase N2B drug delivery during the intranasal administration of domperidone. The drug solution was administered into the nasal cavity via the tracheal catheter. The section of trachea to the lung was also cannulated to maintain spontaneous breathing.

### 2.4. Biological Distribution Study after an Intravenous Dose of [^125^I]IBZM with an Intranasal Domperidone or Saline Dose

All procedures in mice were conducted under isoflurane inhalation anesthesia. Domperidone diluted with saline was intranasally administered at a dose of 2 μg/g body weight and at a volume of 20 μL, using a micro-syringe via the tracheal catheter. Saline was intranasally administered as the control in the same manner. Approximately 200 kBq [^125^I]IBZM were injected into the tail vein of mice at 0, 15, and 30 min after intranasal domperidone administration, or at 0 min after intranasal saline administration (n = 4 at each time point, 16 mice in total) (Figure 2). At 30 min after the intravenous dose of [^125^I]IBZM, cardiac blood was collected under isoflurane inhalation anesthesia, and the animals were euthanized and quickly fractionated into the olfactory bulb, cerebral cortex, striatum, hippocampus, and cerebellum under ice-cold conditions. The wet tissues were then weighed, and the radioactivity in the weighed tissue samples was measured using a gamma counter (AccuFLEX γARC-7010; Hitachi). Data are expressed as the %injected dose per g wet tissue (%ID/g tissue).

### 2.5. SPECT Imaging after an Intravenous Dose of [^123^I]IBZM with an Intranasal or Intravenous Dose of Domperidone

All procedures in mice were conducted under isoflurane inhalation anesthesia (Figure 2). Domperidone diluted with saline was administered intravenously into the tail vein (n = 3; one mouse died) or intranasally (n = 4) using a micro-syringe via the tracheal catheter at a dose of 2 μg/g body weight and volume of 20 μL. As a control, saline was intranasally administered to the mice (n = 5) in the same manner. Furthermore, 14.0 ± 1.6 MBq [^123^I]IBZM diluted with saline was intravenously administered into the tail vein 15 min after the intranasal dose (IN groups) and intravenous dose (IV groups) of domperidone or saline, and SPECT acquisition started immediately after the dose of [^123^I]IBZM, at 5 min/frame for 90 min using a VECTor^+^ system (MILabs, Utrecht, The Netherlands). SPECT images were obtained and reconstructed using the ordered subset expectation maximization method with 16 subsets and 6 iterations, including scatter and attenuation correction. The voxel size was set to 0.8 × 0.8 × 0.8 mm^3^. Post-reconstruction smoothing filtering was applied using a 1.0-mm Gaussian filter. Image displays were obtained using the medical image data analysis software programs Pmod (ver. 3.7, PMOD Technologies LLC, Zurich, Switzerland) and Amide (exe. 1.0.4-1). Time–activity curves (TACs) for the striatum and cerebellum were generated from the SPECT images, and coronal images were displayed as similar section images. Specific binding was calculated by subtracting SUV_mean_ in the cerebellum from that in the striatum. Binding potential (*BP*) and % of receptor occupancy (%RO) using SUV_mean_ in the equilibrium state were calculated in the striatum and cerebellum [14].

### 2.6. Statistical Analysis

Data are presented as means and standard deviation. After normality testing using the Kolmogorov–Smirnov test for statistical analysis of TACs and specific binding, and the Shapiro–Wilk test for analysis of *BP* and %RO, *p* values were calculated using Welch’s *t*-test for comparisons between two groups and analysis of variance. Tukey’s test was used for comparisons among the three groups. All analyses were conducted using GraphPad Prism 8 statistical software (GraphPad Software, Inc., La Jolla, CA, USA). A *P* value of less than 0.05 was considered indicative of a significant difference.

## 3. Results

The appropriate intranasal dose schedule of domperidone was assessed prior to the SPECT study. In the biological distribution study, we decided to sacrifice the mice 30 min after [^125^I]IBZM injection because the ratio of striatum to cerebellum was highest in each brain tissue 30 min after an intravenous dose of [^125^I]IBZM.

Table 1 shows the biological distributions (%ID/g) for each brain tissue and organ. Compared with the controls that received saline instead of domperidone, a significant decrease in accumulation of [^125^I]IBZM was observed only for the striatum, which expresses high levels of D2R [15], in the group that was dosed with intranasal domperidone at 15 min before the [^125^I]IBZM dose. This result suggested that the dose schedule of intranasal domperidone at 15 min before the [^125^I]IBZM dose was the most suitable for use in the SPECT study to evaluate whether intranasal domperidone has a competitive inhibition effect on striatal D2R binding by [^125^I]IBZM.

Figure 3 shows SPECT images of the brain obtained approximately 30 min after a dose of intravenous [^123^I]IBZM with a dose of intranasal saline (control), in the IV group, and in the IN group. One mouse in the IV group died during SPECT imaging. Compared with the strong accumulation of [^123^I]IBZM seen in the striatum in the control image, there was less accumulation in the IV group, and very little accumulation in the IN group. This result corresponds closely to the result of the [^125^I]IBZM biological distribution study.

Figure 4 shows mean TACs of [^123^I]IBZM for the striatum and cerebellum in the three groups. For the cerebellum, there was no significant difference in SUV_mean_ values at any time point among the groups. For the striatum, there was no significant difference in SUV_mean_ values at any time point between the control and IV groups, whereas SUV_mean_ values in approximately the first 10 min were significantly lower in the IN group than in the other two groups.

Figure 5 shows the specific binding value–time curves for the striatum in the three groups, which were calculated using the SUV_mean_ values for the striatum and cerebellum. Specific binding value at approximately the first 20 min was significantly lower in the IN group compared with the other two groups. The equilibrium state of the specific binding value–time curve for the striatum was achieved at 25–35 min in all three groups. The mean *BP* values in the equilibrium state were 1.27, 1.08, and 0.67 for the control group, IV group, and IN group, respectively; the *BP* value was significantly lower in the IN group than in the other two groups (Table 2). The mean %RO value was significantly higher in the IN group than in the IV group (Table 2).

## 4. Discussion

In the present study, a SPECT imaging method was demonstrated in mice to indirectly evaluate whether temporal accumulation of drugs with an affinity for P-gp occurred in the brain via N2B drug delivery. Domperidone, a D2R ligand that is well-known to be poorly BBB permeable, was used as the model drug for N2B delivery. The mouse animal model was appropriate for N2B delivery because the mouse nose has a high ratio of olfactory region to nasal mucosa, and a surgical procedure was performed to place the tip of a catheter in the nasal cavity for intranasal drug administration. A rodent study is appropriate for new drug screening tests by N2B delivery for the reasons of cost-effectiveness and convenience of use in comparison with nonhuman primates and other animals. It is important to synthesize radioactive ligands with high radioactive concentration to ensure a small injection volume. The radioactive concentration of [^125/123^I]IBZM used in this study was very high. Based on the biological distribution data of [^125^I]IBZM (Table 1), the schedule for the dosing of intranasal domperidone for the SPECT study was set at 15 min before the intravenous [^123^I]IBZM. The significantly lower accumulation of [^125^I]IBZM in the striatum alone in the pretreatment group with intranasal domperidone 15 min before the intravenous [^125^I]IBZM compared with the control indicates the binding of domperidone to D2R in the striatum. This result corresponds closely to the result of the SPECT image analysis for the intranasal domperidone group, in which the accumulation of [^123^I]IBZM in the striatum was lower than that in the intranasal saline group. Conversely, the lack of a significant decrease in [^125^I]IBZM accumulation in the striatum for the pretreatment group of 30 min with intranasal domperidone compared with the control may mean that the distribution of domperidone in the brain via the N2B route is transient due to complex factors, including the effects of P-gp expressed in endothelial cells lining the brain capillaries, as well as the olfactory mucosa [16,17]. In support of this hypothesis, a recent similar study was conducted to assess the N2B delivery of quinidine as a poor BBB-permeable drug in rats. This experiment demonstrated the possibility that quinidine, a P-gp substrate, could be transported into the striatum by intranasal administration in a pretreatment with the P-gp specific inhibitor [18].

The transport pathways in N2B drug delivery include the intracellular route (axonal transport), the paracellular route, the transcellular route through olfactory epithelium, and the trigeminal nerve route through respiratory epithelium [19,20]. Although we could not clarify the detailed mechanism of N2B delivery of domperidone in this study, we consider that the main pathway for nasal transport of domperidone is the paracellular route through the lamina propria of olfactory epithelium. In fact, fluorophore-conjugated dextran, a non-directional tracer, has been shown to be easily and quickly transported into the lamina propria of olfactory epithelium [21,22]. However, further experiments are necessary to clarify in more detail the mechanism of transport of domperidone into the brain.

In the present study, TACs (Figure 4) and specific binding value–time curves (Figure 5) of [^123^I]IBZM for the striatum were significantly lower in the intranasal domperidone group compared with the intranasal saline and intravenous domperidone groups. In addition, *BP* and %RO values were significantly lower in the intranasal domperidone group compared with the intravenous domperidone group. From the perspective of ligand-receptor binding competition, these results also indicate that the intact (non-labeled) intranasal domperidone could be transported to the brain via the N2B pathway and compete with binding of [^123^I]IBZM in the striatum.

As a limitation, the striatum, which has the highest expression of D2R and the highest accumulation of [^123^I]IBZM SPECT imaging in the mouse brain, was analyzed. Because the brain size is small, and the resolution of SPECT images is low, it was difficult to set volumes of interest for each brain region. Imaging studies using rats and monkeys, which have larger brain areas than mice, are needed. In addition, it is necessary to confirm the dose-dependent increase in %RO of domperidone by IN administration and to confirm detailed brain region-selective %RO changes in rats and primates, which have larger brain sizes than mice, in the future. In addition, whether P-gp substrates other than domperidone, such as amisulpride and sulpiride, can also be evaluated for brain migration needs to be examined.

## 5. Conclusions

Indirect SPECT imaging evaluation using intravenous [^123^I]IBZM administration could evaluate the time-dependent effect of intranasal domperidone in a mouse study and may be a suitable method for use in N2B drug delivery research to explore time-dependent brain uptakes and locus-specific biological distribution of such drugs as the P-gp substrate domperidone and other poorly BBB-permeable agents.

## Figures and Tables

**Figure 1 pharmaceutics-14-01026-f001:**
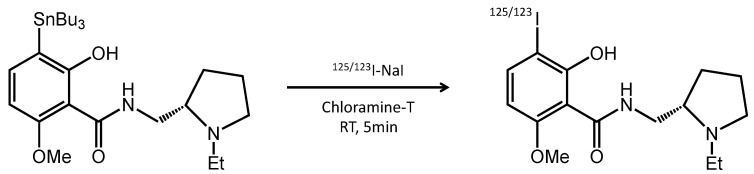
Schematic of ^125/123^I labelling of TBZM. TBZM is labelled with ^125/123^I-NaI by oxidation reaction with chloramine-T for 5 min at room temperature.

**Figure 2 pharmaceutics-14-01026-f002:**
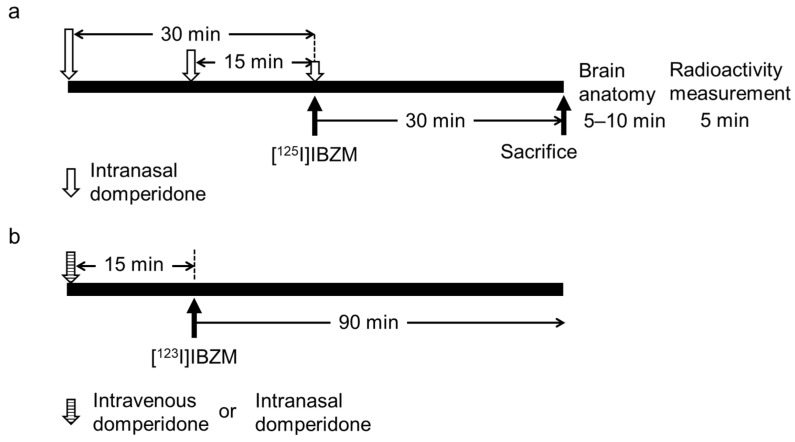
Procedure of the biological distribution study and (**a**) SPECT imaging under isoflurane inhalation anesthesia (**b**) using intravenous [^123^I]IBZM administration with intravenous or intranasal domperidone administration. Normal ddY mice are intravenously or intranasally administered 2 μg/g body weight/20 μL domperidone (control, n = 5; intravenous group, n = 3 (one mouse died); intranasal group, n = 4; for each administration). After mice are sacrificed, brain anatomy is immediately examined, and then radioactivity is measured.

**Figure 3 pharmaceutics-14-01026-f003:**
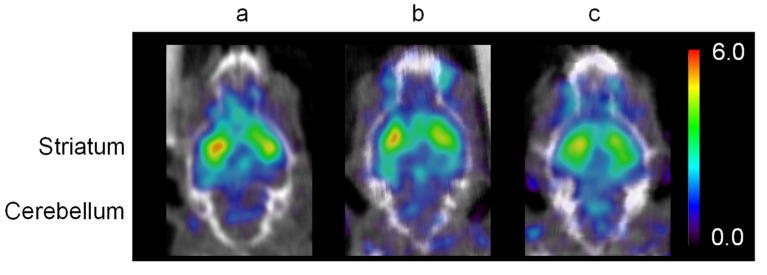
Representative [^123^I]IBZM SPECT images of the control group, (**a**) intravenous domperidone administration (IV group), (**b**) intranasal domperidone administration (IN group), and (**c**) at 25–30 min after intravenous administration of approximately 14 MBq [^123^I]IBZM. There is avid accumulation in the striatum in the control. Accumulation in the striatum is not much lower in the IV group with domperidone compared to the control, but clearly lower in the IN group with domperidone. There is little difference in accumulation in the cerebellum among the control, IV, and IN groups.

**Figure 4 pharmaceutics-14-01026-f004:**
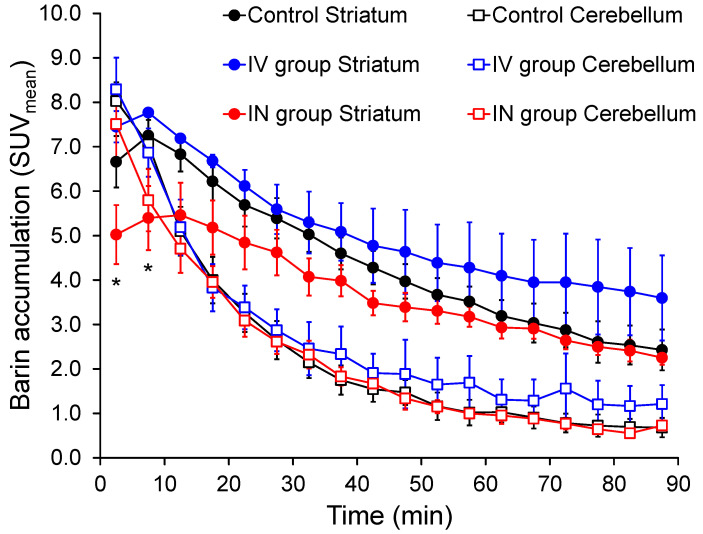
Time–activity curves for the striatum and cerebellum in [^123^I]IBZM SPECT images of the control, IV, and IN groups. In the striatum, there is no significant difference between the control and IV groups with domperidone, but values are significantly lower in the IN group during the first 10 min compared with the control and IV groups. In the cerebellum, there is no significant difference among the groups. * *p* < 0.05 between control and IN group or IV and IN group.

**Figure 5 pharmaceutics-14-01026-f005:**
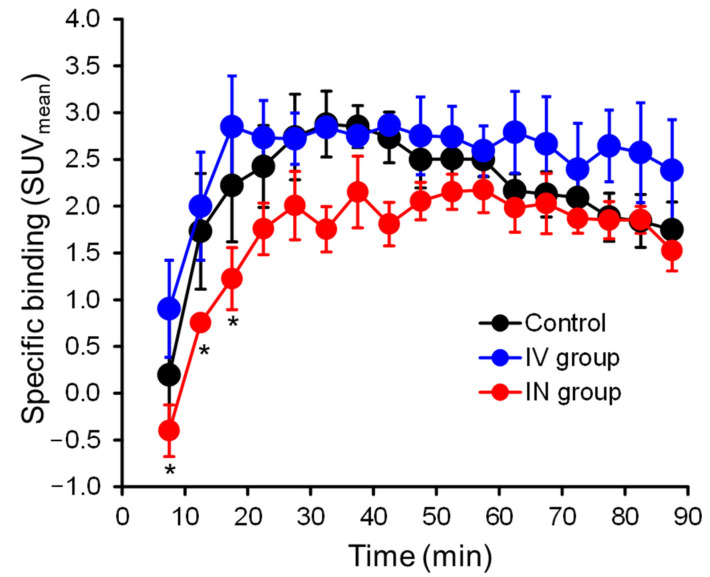
Specific binding of the striatum in the control, IV, and IN groups. Values are significantly lower in the IN group for about 20 min after intravenous [^123^I]IBZM administration compared with the control and IV groups.* *p* < 0.05 between control and IN groups.

**Table 1 pharmaceutics-14-01026-t001:** Biological distribution of [^125^I]IBZM in control mice and following intranasal domperidone administration.

	Accumulation After [^125^I]IBZM Administration (%ID/g)			
Organ	Control Group	IN Groups		
		0 min	15 min	30 min
Blood	2.03 ± 0.28	1.22 ± 0.16	1.69 ± 0.31	1.67 ± 0.21
Olfactory bulb	5.02 ± 1.00	3.56 ± 0.56	3.87 ± 0.70	4.63 ± 1.19
Cerebral cortex	5.07 ± 1.10	4.72 ± 0.57	4.54 ± 0.35	4.59 ± 1.02
Striatum	11.06 ± 3.33	9.27 ± 1.26	6.87 ± 1.95 *	9.92 ± 2.01
Hippocampus	4.37 ± 1.12	4.04 ± 0.54	3.47 ± 0.36	3.83 ± 0.94
Cerebellum	2.97 ± 0.19	3.00 ± 0.14	3.52 ± 0.42	2.90 ± 0.86

%ID/g, percent injected dose per gram of tissue. Values are the mean ± standard deviation obtained from four mice in each group. * *p* < 0.05 compared with control mice.

**Table 2 pharmaceutics-14-01026-t002:** Mean *BP* and %RO values in the equilibrium state in control mice and in the IV and IN groups.

	*BP*	%RO
Control	1.27 ± 0.16	
IV group	1.08 ± 0.11	−4.38 ± 10.95
IN group	0.67 ± 0.08 *	35.63 ± 7.35 *

* *p* < 0.05 compared with control mice or IV group.

## Data Availability

The authors confirm that the data supporting the findings of this study are available within the article.

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
