# Peer review of "Indirect SPECT Imaging Evaluation for Possible Nose-to-Brain Drug Delivery Using a Compound with Poor Blood–Brain Barrier Permeability in Mice"

_pharmaceutics, 2022, doi:10.3390/pharmaceutics14051026_

Round 1

Reviewer 1 Report

The presented paper is, in my opinion, well written and organized with a clear outcome. The findings are interesting and it deserves the publication in high rank journal like this since the article can attract the audience of the journal. I have some minor revisions for further improving the quality and the clarity of the paper

-Abstract IN group. BP and %RO acronyms should be defined in the abstract, and please check through the text to adjust all acronyms

-purity of purchased chemicals should be added considering the information of the manufacturer

-Authors claim that [125/123I]IBZM was synthesized. Although is reported the references, I kindly ask to report the synthetic route adding a Scheme for the synthesis.

Moreover, characterization of the synthesized compound should be added. 1H and 13C NMR   should be reported for the compound and the spectra should be provided as supplementary material

-number of animals should be clearly reported in the materials and Methods section in the Animals. A pure curiosity, authors used male animals why this choose? there is an issue about the gender, please discuss it.

Author Response

Reviewer 1:

The presented paper is, in my opinion, well written and organized with a clear outcome. The findings are interesting and it deserves the publication in high rank journal like this since the article can attract the audience of the journal. I have some minor revisions for further improving the quality and the clarity of the paper

We would like to thank reviewer 1 for the valuable comments on our manuscript. We have carefully read the comments and have revised the manuscript appropriately. The answers to the comments are provided in a point-by-point manner.

-Abstract IN group. BP and %RO acronyms should be defined in the abstract, and please check through the text to adjust all acronyms

Answer: We defined BP and %RO as binding potential and % of receptor occupancy in the abstract as follows:

Binding potential and % of receptor occupancy were significantly lower and higher, respectively, in the IN group than in the control and IV groups.

-purity of purchased chemicals should be added considering the information of the manufacturer

Answer: We added the purity of the chemicals in Materials and methods.

-Authors claim that [125/123I]IBZM was synthesized. Although is reported the references, I kindly ask to report the synthetic route adding a Scheme for the synthesis.

Answer: We added a schematic of [125/123I]IBZM in Figure 1 and the Materials and methods.

Moreover, characterization of the synthesized compound should be added. 1H and 13C NMR should be reported for the compound and the spectra should be provided as supplementary material

Answer: S-(-)-IBZM ([127I]IBZM, stable isotope) was purchased from ABX. They performed 1H-NMR, and the purity was >95%. We provide the spectrum of [125/123I]IBZM in comparison with the spectrum of [127I]IBZM in the HPLC system as supplemental material. In addition, we added the description of the characterization of synthesized [125/123I]IBZM in the Materials and methods, as follows:

The spectrum of [125/123I]IBZM was identified in comparison with the spectrum of [127I]IBZM in the HPLC system (supplemental material).  

-number of animals should be clearly reported in the materials and Methods section in the Animals. A pure curiosity, authors used male animals why this choose? there is an issue about the gender, please discuss it.

Answer: We used 29 mice in total in the biological distribution study (n=16) and the SPECT imaging study (n=13). However, one mouse died in the intravenous domperidone administration group during SPECT imaging. As for sex differences, because the dopaminergic system can be affected by the estrous cycle, males were used to eliminate this effect (Dreher JC, et al. Menstrual cycle phase modulates reward-related neural function in women. Proc Natl Acad Sci U S A. 2007;104(7):2465-70.)

Reviewer 2 Report

The authors have made an interesting paper about SPECT imaging with intra nasal drugs. This manuscript is not ready for publication in Pharmaceutics. Nevertheless, there are some comments that the authors need to address.

  1. The discussion section is weak, not commenting about the other regions of the brain that were evaluated, as well as any clear advantage of its presence in the striatum.

Furthermore, the benefits of less accumulation in the abstract should not be mentioned.

  1. Lines 52-53. Authors should explain this sentence more clearly.
  2. Lines 63-66. Including a reference that support this state.
  3. Figure 1a should be more detailed.
  4. Figure 1. n=3 in the intravenous group is too small to obtain consistent results. Experiments should be repeated by increasing the number of animals.
  5. Figure 4 shows significant differences at 30, 40 and 50 minutes. Therefore, the authors should provide additional clarification on this point.

Author Response

Reviewer 2:

The authors have made an interesting paper about SPECT imaging with intra nasal drugs. This manuscript is not ready for publication in Pharmaceutics. Nevertheless, there are some comments that the authors need to address.

We would like to thank reviewer 2 for the valuable comments on our manuscript. We have carefully read the comments, and we have revised the manuscript appropriately. Our answers to the comments are presented in a point-by-point manner below.

The discussion section is weak, not commenting about the other regions of the brain that were evaluated, as well as any clear advantage of its presence in the striatum.

Furthermore, the benefits of less accumulation in the abstract should not be mentioned.

Answer: We revised the limitations of this study in the Discussion, as follows:

“As a limitation, the striatum, which has the highest expression of D2R and the highest accumulation of [123I]IBZM SPECT imaging in the mouse brain, was analyzed. Because the brain size is small, and the resolution of SPECT images is low, it was difficult to set volumes of interest for each brain region. Imaging studies using rats and monkeys, which have larger brain areas than mice, are needed.” In the abstract, we added “Thus, brain-migrated domperidone inhibited D2R binding of [123I]IBZM.”.

Lines 52-53. Authors should explain this sentence more clearly.

Answer: Neotrofin, a P-gp substrate, administered intracerebroventricularly in the lateral cerebral ventricle, is inhibited from being excreted from the brain when combined with verapamil, a P-gp inhibitor. P-gp substrate drugs like Neotrofin may temporarily accumulate in the brain via N2B drug delivery because they are excreted from the brain when they are recognized by P-gp in the brain.

Lines 63-66. Including a reference that support this state.

Answer: We cited reference 4 (5) in lines 55 to 66. Therefore, reference 4 (5) was moved from line 59 to line 66.

Figure 1a should be more detailed.

We added “brain anatomy” and “radioactivity measurement” in Figure 1a and the following figure legend: “After mice are sacrificed, brain anatomy is immediately examined, and then the radioactivity is measured.” 

Figure 1. n=3 in the intravenous group is too small to obtain consistent results. Experiments should be repeated by increasing the number of animals.

Answer: Actually, we used 4 mice in the intravenous group, but one mouse died because rapidly elevated blood levels of domperidone may cause QT prolongation and induce ventricular tachycardia and fatal arrhythmia [Domperidone and long QT syndrome Curr Drug Saf. 2010;5(3):257-62]. In Europe, a human death occurred with intravenous domperidone, and intravenous domperidone was withdrawn from the market. In this study, the intravenous domperidone group was less important than the intranasal domperidone group and control. We also have to consider the animal ethics standpoint. Therefore, additional mouse experiments are difficult.

Figure 4 shows significant differences at 30, 40 and 50 minutes. Therefore, the authors should provide additional clarification on this point.

Answer: There were no significant differences between control and IN (P=0.0522) at about 30 minutes, (P=0.1334) at about 40 minutes, and (P=0.7412) at about 50 minutes using Tukey’s test in GraphPad Prism 8 statistical software. Tukey’s test is generally considered a robust statistical technique.

Reviewer 3 Report

Excellent work that experimentally validates a method to evaluate the efficacy of the N2B pathway for molecules that are unable to overcome the BBB, certainly deserves to be published by Pharmaceutics.

Minor remarks:

  • The acronyms BP and RO% must be defined also in the abstract.
  • In the paragraph of labeling of IBZM, define the salt used to adjust pH and indicate the exact concentration; e.g., 700 mcl of sodium hydrogen carbonate ... M.

Major revision:

Authors should comment in the Discussion about the need to validate this method by dosing domperidone into brain tissue by a method, e.g. chemical, in a subsequent study.

Author Response

Reviewer 3:

Excellent work that experimentally validates a method to evaluate the efficacy of the N2B pathway for molecules that are unable to overcome the BBB, certainly deserves to be published by Pharmaceutics.

We would like to thank reviewer 3 for the valuable comments on our manuscript. We have carefully read the comments, and we have revised the manuscript appropriately. Our answers to the comments are provided in a point-by-point manner below.

Minor remarks:

The acronyms BP and RO% must be defined also in the abstract.

Answer: We defined BP and %RO as binding potential and % of receptor occupancy in the abstract as follows:

Binding potential and % of receptor occupancy were significantly lower and higher, respectively, in the IN group than in the control and IV groups.

In the paragraph of labeling of IBZM, define the salt used to adjust pH and indicate the exact concentration; e.g., 700 mcl of sodium hydrogen carbonate ... M.

Answer: A total of 700 ml of 0.12 M sodium hydrogen carbonate were used (Materials and methods).

Major revision:

Authors should comment in the Discussion about the need to validate this method by dosing domperidone into brain tissue by a method, e.g. chemical, in a subsequent study.

Answer: We added the following as a limitation in the Discussion.

“As a limitation, the striatum, which has the highest expression of D2R and the highest accumulation of [123I]IBZM SPECT imaging in the mouse brain, was analyzed. Because the brain size is small, and the resolution of SPECT images is low, it was difficult to set volumes of interest for each brain region. Imaging studies using rats and monkeys, which have larger brain areas than mice, are needed. In addition, it is necessary to confirm the dose-dependent increase in %RO of domperidone by IN administration and to confirm detailed brain region-selective %RO changes in rats and primates, which have larger brain sizes than mice, in the future. In addition, whether P-gp substrates other than domperidone, such as amisulpride and sulpiride, can also be evaluated for brain migration needs to be examined.”
